# A Benchmark for Systematic Generalization in Grounded Language Understanding

**Laura Ruis**[*]
University of Amsterdam
laura.ruis@student.uva.nl

**Jacob Andreas**
Massachusetts Institute of Technology
jda@mit.edu

**Marco Baroni**
ICREA
Facebook AI Research
mbaroni@fb.com

**Diane Bouchacourt**
Facebook AI Research
dianeb@fb.com

**Brenden M. Lake**
New York University
Facebook AI Research
brenden@nyu.edu

## Abstract

Humans easily interpret expressions that describe unfamiliar situations composed from familiar parts ("greet the pink brontosaurus by the ferris wheel"). Modern neural networks, by contrast, struggle to interpret novel compositions. In this paper, we introduce a new benchmark, gSCAN, for evaluating compositional generalization in situated language understanding. Going beyond a related benchmark that focused on syntactic aspects of generalization, gSCAN defines a language *grounded* in the states of a grid world, facilitating novel evaluations of acquiring linguistically motivated rules. For example, agents must understand how adjectives such as 'small' are interpreted relative to the current world state or how adverbs such as 'cautiously' combine with new verbs. We test a strong multi-modal baseline model and a state-of-the-art compositional method finding that, in most cases, they fail dramatically when generalization requires systematic compositional rules.

## 1   Introduction

Human language is a fabulous tool for generalization. If you know the meaning of the word 'small', you can probably pick the 'small wampimuk' among larger ones, even if this is your first encounter with wampimuks. If you know how to 'walk cautiously,' you can infer how to 'bike cautiously' through a busy intersection (see example in Figure 1). The ability to learn new words from limited data and use them in a variety of contexts can be attributed to our aptness for systematic compositionality [9, 33]: the algebraic capacity to understand and produce potentially infinite combinations from known components. Modern deep neural networks, while strong in many domains [29], have not mastered comparable language-based generalization challenges, a fact conjectured to underlie their sample inefficiency and inflexibility [26, 25, 8]. Recent benchmarks have been proposed for language-based generalization in deep networks [20, 16, 17], but they do not specifically test for a model's ability to perform rule-based generalization, or do so only in limited contexts. Systematic, rule-based generalization is instead at the core of the recently introduced SCAN dataset [25] (see [19, 3] for related ideas). In a series of studies, Lake, Baroni and colleagues [5, 30, 10] tested various standard deep architectures for their ability to extract general composition rules supporting zero-shot interpretation of new composite linguistic expressions (can you tell what 'dax twice' means, if you know the meaning of 'dax' and 'run twice'?). In most cases, neural networks were unable to generalize correctly. Very recent work has shown that specific architectural or training-regime adaptations allow

---

[*]Work done during an internship at Facebook Artificial Intelligence Research.

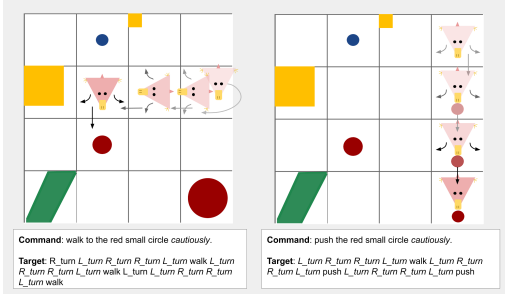

Figure 1: gSCAN evaluates context sensitivity in situated language understanding. In these two simplified examples, the same determiner phrase 'the red small circle' has different referents and demands different action sequences. Being cautious means looking both ways ('*L_turn R_turn R_turn L_turn*') before each step.

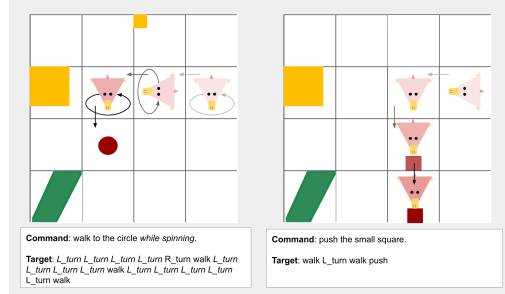

Figure 2: Examples showing how to 'walk while spinning' and how to 'push.' On the left, the agent needs to spin around ('*L_turn L_turn L_turn L_turn*') before it moves by one grid cell. On the right, it needs to push a square all the way to the wall.

deep networks to handle at least some of the SCAN challenges [2, 27, 34, 36, 14]. However, it is unclear to what extent these proposals account for genuine compositional generalization, and to what extent they are 'overfitting' to the limitations of SCAN.

SCAN simulates a navigation environment through an interpretation function that associates linguistic commands ('walk left') to sequences of primitive actions (*L_turn walk*). SCAN, however, is not *grounded*, in that it lacks a 'world' with respect to which commands are interpreted: instead the agent must simply associate linguistic strings with fixed sequences of action symbols, mapping syntactic strings (word sequences) to other syntactic strings (action label sequences). In real languages, by contrast, the process by which utterances are understood is both compositional and *contextual*: references to entities and descriptions of actions must be interpreted with respect to a particular state of the world. The interaction between compositionality and context introduces new types of generalization an intelligent agent might have to perform. For example, consider the meaning of size adjectives such as 'small' and 'large'. The determiner phrases 'the small bottle' and 'the large bottle' might refer to the same bottle, depending on the sizes of the surrounding bottles. We wonder whether this and related notions of compositional generalization can be addressed using existing techniques, but SCAN's context insensitivity makes it impossible to investigate broader notions of generalization.

We introduce *grounded* SCAN (gSCAN), a new benchmark that, like the original SCAN, focuses on rule-based generalization, but where meaning is grounded in states of a grid world accessible to the agent. This allows gSCAN to evaluate eight types of compositional generalization (mostly from a single training set), whereas most benchmarks focus on just one or two types. For example, Figure 1 illustrates context-sensitivity in compositionality: how the target referent 'the red small circle', and the action sequence required to navigate there, will change based on the state of the world. It also illustrates modification-based compositionality: once learning how to walk somewhere 'cautiously' (Figure 1 left), can models walk elsewhere cautiously or push an object cautiously (Figure 1 right)? On these and other generalizations, we test a baseline multi-modal model representative of contemporary deep neural architectures, as well as a recent method proposed to address compositional generalization in the original SCAN dataset (GECA, [2]). Across eight different generalization splits, the baseline dramatically fails on all but one split, and GECA does better on only one more split. These results demonstrate the challenges of accounting for common natural language generalization phenomena with standard neural models, and affirm gSCAN as a fruitful benchmark for developing models with more human-like compositional learning skills.

## 2   Related Work

Recent work has recognized the advantages of compositional generalization for robustness and sample efficiency, and responded by building synthetic environments to evaluate models on aspects of this skill [7, 21, 25, 30, 4, 16, 17, 8, 19, 3]. Several of these works also ground the semantics of language in a different modality. Bahdanau et al. [4] evaluate binary questions about object relations and probe unseen object combinations. Chevalier-Boisvert et al. [8] study curriculum learning in a grid world of navigation-related tasks. The authors show that for tasks with a compositional structure

agents generalize poorly and need large amounts of demonstrations. Crucially, the difference between these works and ours is that we rely on evidence that humans systematically generalize in language [9, 33, 28] and only test for linguistic, rule-based generalization. Hill et al. [17] also test for linguistic generalization and evaluate unseen combinations of verbs and objects in skill learning, showing the benefits of richer grounded environments. Our benchmark includes a similar test for verb-object binding, but adds challenges to more comprehensively cover the multi-faceted nature of systematic compositionality.

Lake and Baroni [25] proposed the SCAN benchmark for evaluating systematic generalization, which was distinguished by testing for linguistic generalization and learning abstract compositional rules (see also [7, 3, 22, 19]). SCAN concerns instructions generated by a phrase-structure grammar that can unambiguously be translated into action sequences by applying an interpretation function. The data is split into training and test sets that contain systematic differences. For example, models must interpret phrases that contain primitives only encountered in isolation at training time (e.g., inferring that the command 'jump twice' translates to actions *jump jump* when you know that 'walk twice' translates to *walk walk* and 'jump' translates to *jump*). SCAN however lacks grounding, which severely limits the variety of linguistic generalizations it can examine.

In addition to adding grounding, gSCAN was developed in ways that render previous SCAN-based methods either inapplicable or unsuccessful. Gordon et al. [14] formalize the compositional skills in SCAN as equivariance to a certain group of permutations. Their model is hard-coded to be equivariant to all permutations of SCAN's verb primitives and succeeds on some of the tasks. However, the method only tackles local permutations in the input command (e.g., swapping 'walk' for 'jump') that result in local permutations in the action sequence (swapping *walk* for *jump*). More realistically, in gSCAN, permuting words can result in referencing different objects, different interactions, or different manners of moving. Permutations in the instructions modify the actions in a non-local manner, in their terminology, a problem that would also affect the recent syntax-semantics separation approach [36]. Another new approach uses meta-learning to succeed on some SCAN splits [27]. This model 'learns how to learn' new primitives in meta-training episodes where words are randomly mapped to meanings. But it is unclear how such episodes should be designed for gSCAN, since there are no random mapping to exploit between primitives and action symbols. Thus, these new methods [14, 36, 27] are inapplicable to gSCAN, at least in their current forms, and may require highly non-trivial extensions to attempt the benchmark.

Good-enough compositional data augmentation (GECA) [2] is a model-agnostic method that obtains good results on SCAN and is applicable to gSCAN. GECA identifies sentence fragments which appear in similar environments and uses those to generate more training examples. For instance, when one infers that *'the cat sang'*, *'the wug sang'* and *'the cat danced'* are high probability training sentences, then *'the wug danced'* is also probable, but *'the sang danced'* is not. The assumption here is that *'cat'* and *'wug'* are interchangeable, and GECA indeed helps in such cases. As our results will show, this assumption is unreliable in more realistic, grounded language understanding.

## 3  The Grounded SCAN Benchmark

We aim to test a broad set of phenomena in situated language understanding where humans should easily generalize, but where we expect computational models to struggle due to the systematicity of the differences between train and test. For a learner agent, the goal is to process a synthetic language command combined with a world state and produce a sequence of target actions that correctly execute the input command in the world state (see Figure 1 and 2), which can be treated as a multi-modal sequence-to-sequence supervised learning task. In this section we describe what tools we work with to design the tests, and in Section 5 we describe in detail how each linguistic phenomenon is tested. The code to generate the benchmark and the data used in the experiments are both publicly available.[2]

**Instructions.** Grounded SCAN (gSCAN) asks agents to execute instructions in a 2D grid world with objects. We build on the formal approach of SCAN [25] while evaluating a much wider range of linguistic generalizations by grounding the semantics of the input instructions. The world model allows us to examine how often an agent needs to see 'move cautiously' before applying 'cautiously' in a novel scenario, whether an agent can identify a novel object by reasoning about its relation to other objects, and whether an agent can infer how to interact with objects by identifying abstract

object properties. Figure 2 shows two example commands and corresponding action sequences, which are simplified but representative of gSCAN (actual gSCAN examples use a larger grid and more objects). On the left, the agent must generate the target actions that lead to the circle 'while spinning.' All adverbial modifiers such as 'cautiously' or 'while spinning' require applying complex, context-sensitive transformations to the target sequence, going beyond the simple substitutions and concatenations representative of SCAN. On the right (Figure 2), the agent must push a small square, where pushing requires moving something as far as possible without hitting the wall (in this case), or another object. The agent can also 'pull' objects, in which case it would pull the object back as far as possible. The full phrase-structure grammar for instructions is provided in Appendix A.

**World model**. Each instruction is paired with a relevant world state, presented to the agent as a tensor $\mathbf{X}_s \in \mathbb{R}^{d \times d \times c}$ for grid size $d$ ($d = 6$ or $12$ depending on split)[3]. The object at each grid cell is defined via one-hot encodings along three property types, namely color $\mathcal{C} = \{$red, green, blue, yellow$\}$, shape $\mathcal{S} = \{$circle, square, cylinder$\}$, and size $\mathcal{D} = \{1, 2, 3, 4\}$. Specifying the agent location and heading requires five more channels, and thus the tensorwidth is $c = 5 + |\mathcal{C}| + |\mathcal{S}| + |\mathcal{D}|$. As for the outputs, agents produce strings composed of action symbols $\{walk, push, pull, stay, L\_turn, R\_turn\}$.

To ensure that only relevant world states are combined with an instruction, each instruction imposes several constraints on combined world states. For instance, each target referent from the instruction determiner phrase is ensured to be unique (only one possible target in "walk to the yellow square"). Moreover, to conform to natural language pragmatics, if a size modifier is used there is always a relevant distractor. For example, when the target referent is 'the small square', we additionally place a square that is larger (Figure 2 right). Appendix B details how objects are placed in the world.

Further, objects of size 1 and 2 are assigned the latent class *light*, and objects of size 3 and 4 are *heavy*. This division determines how the agent should interact with the object. If an object is light, it needs to be pushed once to move it to the next cell, executed by the action command 'push' (similarly for 'pull'). If an object is heavy, it needs to be pushed twice to move to the next cell ('push push').

**Data splits**. Equipped with this framework, we design splits with systematic differences between training and test. We distinguish two broad types of tests, compositional generalization and length generalization. To facilitate more rapid progress and lower compute requirements, we designed the eight systematic splits to require training just two models—one for compositional and one for length generalization—as multiple tests use the same training set. For both broad types of tests we also examine a 'random split' with no systematic differences between training and test to ensure that the agents are able to execute commands to a high accuracy when the they require no systematic compositionality (Section 5A).

'Compositional generalization' evaluates combining known concepts into novel meaning (detailed in Section 5B-H). From a single training set we can evaluate how an agent handles a range of systematic generalizations, including novel object property combinations ('red square'; Section 5B,C), novel directions (a target to the south west; 5D), novel contextual references ('small yellow circle'; 5E), and novel adverbs ('pull cautiously'; 5G,H). For example, we model an analogue of the 'wampimuk' case from the introduction by holding out all examples where a circle of size 2 is referred to as 'the small circle' (Section 5E). We test whether models can successfully pick out the small circle among larger ones, even though that particular circle is referred to during training only as 'the circle' (with no other circles present) or 'the large circle' (with only smaller circles present). The shared training set across splits has more than 300k demonstrations of instructions and their action sequences, and each test instruction evaluates just one systematic difference. For more details on the number of examples in the training and test sets of the experiments, refer to Appendix C. To substantiate the fairness of the tests we also discuss the number and combinations of concepts available during training [13] (see individual splits in Section 5).

'Length generalization' (Section 5I) evaluates a persistent problem with sequence generation models: generalizing beyond the lengths seen during training [15]. Since length generalization is entirely unsolved for SCAN, even for methods that made progress on the other splits [25, 5, 36, 14] , we also separately generate a split to evaluate generalization to longer action sequences. We do this by using a larger grid size than in the compositional generalization splits ($d = 12$), and hold out all examples with target sequences of length $m > 15$. During training, a model sees all the possible instructions (see Appendix C), but they require fewer actions than at test time (up to $m = 47$).

Table 1: Results for each split, showing exact match accuracy (average of 3 runs ± std. dev.). Models fail on all splits except A, C, and F.

| Split | Exact Match (%) | |
|---|---|---|
| | Baseline | GECA |
| A: Random | 97.69 ± 0.22 | 87.6 ± 1.19 |
| B: Yellow squares | 54.96 ± 39.39 | 34.92 ± 39.30 |
| C: Red squares | 23.51 ± 21.82 | 78.77 ± 6.63 |
| D: Novel direction | 0.00 ± 0.00 | 0.00 ± 0.00 |
| E: Relativity | 35.02 ± 2.35 | 33.19 ± 3.69 |
| F: Class inference | 92.52 ± 6.75 | 85.99 ± 0.85 |
| G: Adverb $k = 1$ | 0.00 ± 0.00 | 0.00 ± 0.00 |
| Adverb $k = 5$ | 0.47 ± 0.14 | - |
| Adverb $k = 10$ | 2.04 ± 0.95 | - |
| Adverb $k = 50$ | 4.63 ± 2.08 | - |
| H: Adverb to verb | 22.70 ± 4.59 | 11.83 ± 0.31 |
| I: Length | 2.10 ± 0.05 | - |

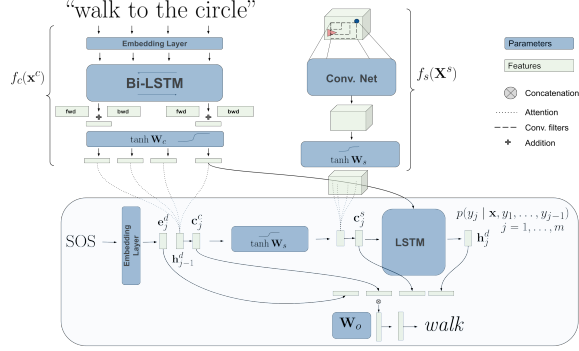

Figure 3: Baseline network. The command encoder is a biLSTM ($f_c$; top left), and the world state encoder is a CNN ($f_s$; top right). A LSTM decoder jointly attends over $f_c$ and $f_s$ to produce action sequences (bottom). 'SOS' is the start-of-sequence token that kicks off generation.

## 4 Baselines

Models are trained using supervised learning to map instructions to action sequences, given a world context. We train a multi-modal neural baseline to generate action sequences, conditioned on the input commands and world state (Figure 3). The architecture is not new and uses standard machinery, e.g., [32], but we explain the key components below for completeness (details in Appendix D).

The baseline is a sequence-to-sequence (seq2seq) [38] model fused with a visual encoder. It uses a recurrent 'command encoder' to process the instructions ("walk to the circle" in Figure 3) and a 'state encoder' to process the grid world. A recurrent decoder generates an action sequence (e.g., *walk*) through joint attention over the command steps and grid cells. The input tuple $\mathbf{x} = (\mathbf{x}^c, \mathbf{X}^s)$ includes the command sequence $\mathbf{x}^c = \{x_1^c, \ldots, x_n^c\}$ and the world state $\mathbf{X}^s \in \mathbb{R}^{d \times d \times c}$, for a $d \times d$ grid. The target sequence $\mathbf{y} = \{y_1, \ldots, y_m\}$ is modeled as $p_\theta(\mathbf{y} \mid \mathbf{x}) = \prod_{j=1}^{m} p_\theta(y_j \mid \mathbf{x}, y_1, \ldots, y_{j-1})$.

**Command encoder**. The network processes the instruction with a bidirectional LSTM [18, 37] denoted $\mathbf{h}^c = f_c(\mathbf{x}^c)$ (Figure 3). It produces $\mathbf{h}^c = \{h_1^c, \ldots, h_n^c\}$ with a vector for each of the $n$ words.

**State encoder**. The network perceives the initial world state through a convolutional network (CNN; Figure 3) denoted $\mathbf{H}^s = f_s(\mathbf{X}^s)$, with three kernel sizes [40]. It produces a grid-based representation of the world state $\mathbf{H}^s \in \mathbb{R}^{d \times d \times 3c_{\text{out}}}$ with $c_{\text{out}}$ as the number of feature maps per kernel size.

**Decoder.** The output decoder $f_d$ models the action sequences given the decoder messages, $p(\mathbf{y} | \mathbf{h}^c, \mathbf{H}^s)$. At each step, the previous output $y_{j-1}$ is embedded $\mathbf{e}_j^d \in \mathbb{R}_e^d$, leading to $\mathbf{h}_j^d = \text{LSTM}([\mathbf{e}_j^d; \mathbf{c}_j^c; \mathbf{c}_j^s], \mathbf{h}_{j-1}^d)$. Context vectors $\mathbf{c}_j^c$ and $\mathbf{c}_j^s$ use double attention [11]. First, the command context is $\mathbf{c}_j^c = \text{Attention}(\mathbf{h}_{j-1}^d, \mathbf{h}^c)$, attending over the input steps and producing a weighted average of $\mathbf{h}^c$ (Appendix D for definition). Second, conditioning on $\mathbf{c}_j^c$, the state context is $\mathbf{c}_j^s = \text{Attention}([\mathbf{c}_j^c; \mathbf{h}_{j-1}^d], \mathbf{H}^s)$, attending over grid locations and producing a weighted average of $\mathbf{H}^s$. The action emission $y_j$ is then $p(y_j \mid \mathbf{x}, y_1, \ldots, y_{j-1}) = \text{softmax}(\mathbf{W}_o[\mathbf{e}_j^d; \mathbf{h}_j^d; \mathbf{c}_j^c; \mathbf{c}_j^s])$.

**Training.** Training optimizes cross-entropy using Adam with default parameters [23]. Supervision is provided by ground-truth target sequences, with the convention of traveling horizontally first and then vertically (either is okay at test). The learning rate starts at $0.001$ and decays by $0.9$ every $20,000$ steps. We train for $200,000$ steps with batch size $200$. The best model was chosen based on a small development set of $2,000$ examples (full details in Appendix E). The most important parameters were the kernel sizes, chosen as $1, 5$, and $7$ for 6x6 states and $1, 5$, and $13$ for 12x12 states.

**Good-enough compositional data augmentation.** A GECA-enhanced model was run on gSCAN with similar parameters to [1]. The context windows are full sentences, with a gap size of one and a maximum of two gaps (Appendix E for details). GECA receives an input sequence consisting of the natural language command concatenated with an output sequence containing a linearized

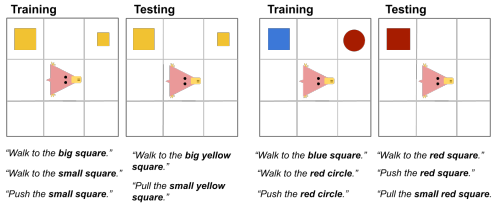
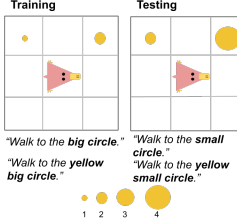
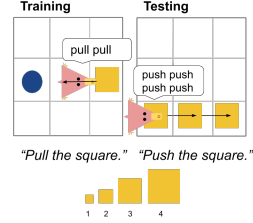

Figure 4: Generalizing from calling an object "big square" to calling it 'big yellow square' (left), and from 'red' and 'square' to 'red square' (right).

Figure 5: Generalizing from calling an object "big" to calling it "small."

Figure 6: Generalizing from pulling to pushing a heavy square.

representation of the target object's feature vector. After generating augmented sequences, we re-apply these to the gSCAN dataset by modifying training examples with augmentable input sentences. Modification leaves action sequences unchanged while changing the commands and environment features. For example, command "walk to a red circle" could become "walk to a red square", and the world state would analogously replace the target red circle with a red square.

## 5 Experiments

Our main contribution is the design of test sets that require different forms of linguistic generalization. We trained the models from the previous section on the two training sets described in Section 3. A summary of results is shown in Table 1, with each split detailed below. All experiment and model code is available so our results can be reproduced and built upon.[4]

**A: Random split**. The random split verifies that the models can learn to follow gSCAN commands when there are no systematic differences between training and test. The training set covers all instructions that appear in this split, each coupled with more than 170 unique world states (Appendix C for more details), meaning that test examples only differ in the world states that are combined with the instruction (e.g., the training set might instruct the agent to "walk to the small red circle" with a target red circle in row 2 and column 3, and at test time the agent needs to generalize to walking to the small red circle in row 1 column 4). The baseline model achieves near perfect exact match accuracy ($97.69\% \pm 0.22$ mean over 3 runs, reported with standard deviation) on the 19,282 test examples, where exact match means that the entire action sequence is produced correctly. GECA performs worse ($87.6\% \pm 1.19$), which is unsurprising since the assumption underlying the data augmentation procedure in GECA is that phrases that appear in similar environments can be permuted. This is not always correct in gSCAN (none of the verbs and adverbs can be permuted).

**B, C: Novel composition of object properties.** Here we examine a well-studied but still challenging type of compositionality [20, 16, 12]: whether a model can learn to recombine familiar colors and shapes to recognize a novel color-shape combination (see Figure 4 for examples). Given that objects are clearly distinguished from the referring expressions denoting them (a red square of size 2 may have referents 'square', 'red square', 'small red square', or 'large red square'), we consider two separate setups, one involving *composition of references* and another involving *composition of attributes*.

For a first split, we hold out all data examples where a yellow square (of any size) is the target object and is referred to with the determiner phrases 'the yellow square', 'the small yellow square' or 'the big yellow square' (i.e., any phrase containing the color adjective and the shape). The training set contains examples with yellow squares as a target, but they are always referred to without a color: 'the square', 'the big square', or 'the small square', meaning the methods cannot

Table 2: Exact match broken down by referred target (i.e., the target object denoted in the determiner phrase of the instruction). The ⋆ column indicates chance performance of choosing an object uniformly at random and correctly navigating to it.

| Referred Target | ⋆ | Baseline | GECA |
|---|---|---|---|
| 'small red square' | 8.33 | $13.09 \pm 14.07$ | $78.64 \pm 1.10$ |
| 'big red square' | 8.33 | $11.03 \pm 10.29$ | $77.88 \pm 0.95$ |
| 'red square' | 16.67 | $8.48 \pm 0.90$ | $97.95 \pm 0.14$ |
| 'small square' | 8.33 | $27.13 \pm 41.38$ | $66.26 \pm 13.60$ |
| 'big square' | 8.33 | $22.96 \pm 32.20$ | $68.09 \pm 20.90$ |
| 'square' | 50 | $52.92 \pm 36.81$ | $95.09 \pm 7.42$ |

[4]https://github.com/LauraRuis/multimodal_seq2seq_gSCAN

ground that target to the reference 'yellow square' (Figure 4 left). At test time, the model needs to zero-shot generalize to the yellow square being referred to by its color. The second split never sees examples where a red square is the target in training, meaning the methods, in addition to never encountering the determiner phrase 'the red square', cannot ground a reference to this object (Figure 4 right). However, the red square is familiar since it appears often as a non-target background object. There is ample opportunity to learn these color-shape combinations during training: yellow squares are the target reference 16,725 times, and 'square' and 'red' appear in many other contexts.

The baseline shows poor performance on the 'red squares'-split that requires zero-shot generalization to the target red square ($23.51\% \pm 21.82$; again exact match with standard deviation over 3 runs). GECA does substantially better on this split ($78.77\% \pm 6.63$). This is precisely what GECA is designed for; permuting 'red circle' and 'yellow square' during training gives familiarity with 'red square'. Surprisingly, GECA does not improve over the baseline for the 'yellow square'-split ($34.92\% \pm 39.30$ for GECA and $54.96\% \pm 39.39$ for baseline). In this split, yellow squares have been seen during training as a target object, yet never referred to using their color: seeing '(small,big) square' but never '(small,big) yellow square' in the commands. We hypothesize that models overfit this pattern, and while GECA should help by generating instructions including 'yellow', their number is still too small compared to the ones of the form '(small,big) square'. In Appendix C we take a closer look at the differences in the training evidence seen by the baseline model and GECA.

When analyzing accuracy per referred target for the 'red squares'-split, we can reason about what happens (Table 2). The baseline model never sees the referred target 'red square' (except as a background object) and is unable to compose a meaningful representation of this object, confirming past work [24, 30]. It has seen plenty evidence for other red objects (circles and cylinders) and other colored squares (blue or green). Higher performance when only the shape is explicitly denoted (e.g. "walk to the square", when the square happens to be red) is expected because there are at most 2 objects in the world when only shape is denoted (chance is 50%). GECA does well when 'the red square' or 'the square' is used, but is thrown off when a size adjective is mentioned. Again, it seems these adjectives are too strongly grounded to the objects that do appear during training.

Last, we examine if the errors come from target identification or sequence generation. In the random split (A), the agent attends maximally (averaged over steps) to the target object in 94% of episodes. In contrast, in the 'yellow square'- and 'red square'-split, the agent does so in only 0.08% and 47% of error episodes, respectively, showing clear failures of compositional target identification.

**D: Novel direction.** In the next experiment we examine generalizing to navigation in a novel direction. We hold out all examples from the training set where the target object is located to the south-west of the agent. The agent can train on walking in any other direction, and needs to generalize to walking to the west and then the south, or vice-versa. Conceptually, at test time the agent needs to combine the familiar notions of walking to the south and west. Results are in the row 'novel direction' of Table 1. Both methods obtain 0 exact matches over all runs (0% correct). A closer analysis of the predictions (see Figure 2 Appendix F for 2 examples) shows that the agent usually walks all the way west (or south) and then fails to turn to the target object. The attention shows that the agent knows where to go (by attending to the correct grid cell), just not how to get there. Even though there is catastrophic failure at the task overall, the agent learned by the baseline model ends up in the correct row or column of the target $63.10\% \pm 3.90$ of the times, and the agent learned with GECA $58.85\% \pm 3.45$ of the times. This further substantiates that they often walk all the way west, but then fail to travel the distance left to the south, or vice-versa. The results on this split indicate that apparently the methods completely fail to generate target sequences that have either three occurrences of *L_turn* (needed to walk to the west and then south for an agent that starts facing east) or two occurrences of *R_turn* (needed to walk to the south and then west) spread over the target sequence.

**E: Novel contextual references.** In natural language, many words can only be grounded to relative concepts. Which object one refers to when saying 'the small circle' depends on the other circles in the world state. We investigate whether a model can grasp relativity in language by considering a scenario where objects of a specific size (size 2) are never targets correctly picked by the 'small' modifier during training (see Figure 5). At test time, the target is a circle of size 2, which is being correctly referred to as a 'small circle' (the determiner phrase may also mention color). In other words, we hold out for testing all world states where the circle of size 2 is the target and the smallest circle in the world, paired with an instruction containing the word 'small'. The agent can ground size 2 circles to references like 'green circle' or 'big circle' (there are 29,966 such training examples), but

needs to generalize to that same circle being referred to as 'the small circle' at test time. To do this correctly the agent cannot simply memorize how 'small' is used in training, but needs to understand that it's meaning is relative with respect to the world state.

Both methods are again substantially worse than on the random split: $35.2\% \pm 2.35$ for the baseline and $33.19\% \pm 3.69$ for GECA. When breaking down the exact match per referred target it seems like the model is exploiting the fact that when in addition to the size modifier 'small' the color of the circle is specified, it can randomly choose between 2 circles of the specified color, as opposed to randomly choosing between any circle in the world. When generating world states for instructions containing some combination of a color, size modifier and shape in the determiner phrase (e.g., 'the small red circle') during data generation we always generate 2 differently sized objects of each color-shape pair. So when you recognize the color and shape in the instruction, you have a 50% chance of picking the right object. Then how to interact with it if necessary is already familiar. We observe that for data examples where the instruction specifies the color of the target in addition to the size the baseline achieves $53\%.00 \pm 1.36$ and GECA achieves $47.51\% \pm 12.59$, suggesting the agents randomly select a circle of the specified color. As in splits (B) and (C), the model only maximally attends to the correct target in 4% of errors. Thus the obtained performance indicates a complete failure of genuinely understanding 'small' and picking a small circle from among larger ones in arbitrary circumstances.

**F: Novel composition of actions and arguments.** Another important phenomenon in natural language is categorizing words into classes whose entries share semantic properties [35]. We study the simple case of *nominal class inference*, establishing two categories of nouns, that, depending on their weight (they can be light or heavy), will lead to a different interpretation of the verb taking them as patient arguments. Recall from Section 3 that pushing or pulling a heavy object over the same distance (i.e., grid cells) as a light object requires twice as many target actions of 'push' or 'pull'.

This split examines inferences about latent object class and how to correctly interact with objects, as shown in Figure 6. We hold out all examples where the verb in the instruction is 'push', and the target object is a square of size 3, meaning it is in the heavy class and needs to be pushed twice to move by one grid cell. A model should infer that this square of size 3 is 'heavy' from its extensive training experience 'pulling' this object, each time needing two actions to move it (shown through 7,656 training trials). Adding to this experience, all circles and cylinders of this size also needs to be pushed twice (appearing 32,738 times). Note that Hill et al. [17] similarly studies verb-noun binding.

Both methods perform similarly to their accuracy on the random split, namely $92.52\% \pm 6.75$ and $85.99\% \pm 0.85$ by the baseline and GECA respectively (examples with 'push' in the random split were at $96.64\% \pm 0.52$ and for GECA $86.72\% \pm 1.23$), and seem to be able to correctly categorize the square of size 3 in the class heavy and interact with it accordingly. This is consistent with the findings of Hill et al. [17] with regards to generalizing familiar actions to new objects.

**G, H: Novel adverbs.** In the penultimate experiment we look at transforming target sequences in response to how adverbs modify command verbs. The adverbs all require the agent to do something (i.e., generate a particular action sequence) at some predefined interval (see Figures 1 and 2 for examples). To do something *cautiously* means looking both ways before crossing grid lines, to do something *while spinning* requires spinning around after moving to a grid cell, to do something *hesitantly* makes the agent stay put after each step (with the action *stay*), and finally to do something *while zigzagging* only applies to moving diagonally on the grid. Where normally the agent would first travel horizontally all the way and then vertically, when doing something while zigzagging the agent will alternate between moving vertically and horizontally every grid cell.

We design two experiments with adverbs. The first examines learning the adverb 'cautiously' from just one or a few examples and using it in different world states (few-shot learning). For instance, the agent sees examples of instructions with the adverb 'cautiously' during training, and needs to generalize to all possible instructions with that adverb (discarding those with longer target sequences than seen during training). To give the models a fair chance, we examine learning from one to fifty examples. The second experiment examines whether a model can generalize a familiar adverb to a familiar verb, namely 'while spinning' to 'pull'. In this experiment the agent sees ample evidence of both the tested adverb (66,229 examples) and the verb (68,700 examples), but has never encountered them together during training. These experiments are related to the 'around right'-split introduced in SCAN by Loula et al. [30], but in this case, the grounded meaning of each adverb in the world state changes its effect on the target sequence. Therefore we expect methods like the equivariance permutations of Gordon et al. [14], but also GECA, to have no impact on generalization.

During few-shot learning, the models fail catastrophically when learning 'cautiously' from just one demonstration (0% correct; exact match again). We experiment with increasing the number ($k$) of 'cautiously' examples during training (Table 1), but find it only marginally improves the baseline's abysmal performance. Even with $k = 50$, performance is only 4.6% correct, emphasizing the challenges of acquiring abstract concepts from limited examples. For combining spinning and pulling, both methods also struggle ($22.70\% \pm 4.59$ for baseline and $11.83\% \pm 0.31$ for GECA). In each case, performance drops as a function of target length (Figures 3 and 4 Appendix F).

**I: Novel action sequence lengths.** For this split we only train the baseline model, as data augmentation will not produce longer training examples. When trained on actions sequences of length $\leq 15$, the baseline performs well on held-out examples below this length ($94.98\% \pm 0.1$) but degrades for longer sequences ($2.10\% \pm 0.05$ overall; $19.32\% \pm 0.02$ for length 16; $1.71\% \pm 0.38$ for length 17; below $1\%$ for length $\geq 18$). Unsurprisingly, as with SCAN, baselines struggle with this task.

## 6    Conclusion

The SCAN benchmark has catalyzed new methods for compositional learning [2, 27, 14, 36]. Our results, on a new gSCAN (grounded SCAN) benchmark, suggest these methods largely exploit artifacts in SCAN that are not central to the nature of compositional generalization. gSCAN removes these artifacts by introducing more sophisticated semantics through grounding. We trained strong multi-modal and GECA baselines, finding that both methods fail on the vast majority of splits. Both methods succeed only on inferring object class and using it for interaction (similarly to [17]), while GECA improves only on novel composition of object properties ('red squares'). The complete failure on the subsequent splits show advances are needed in neural architectures for compositional learning. Progress on gSCAN may come from continuing the lines of work that have made progress on SCAN. Meta-learning [27] or equivariance permutation [14] could support compositional generalization, if the types of generalizations examined here can be captured in a meta-training procedure or equivariance definition. For now, at least, applying these methods to gSCAN requires highly non-trivial extensions.

In future work, we plan to extend gSCAN to support reinforcement learning (RL), although certain splits are not straightforward to translate ('pushing', 'spinning', etc.) The benchmark seems demanding enough with supervision, without adding RL's sample complexity issues (e.g., RL seems unlikely to improve few-shot learning or target identification, but it may help with length). gSCAN could use RGB images instead of partially-symbolic state representations, although again it should only add difficulty. We expect progress on gSCAN to translate to more naturalistic settings, e.g., [31, 39, 22, 6], since the issues studied in gSCAN feature prominently in realistic NLU tasks such as teaching autonomous agents by demonstration. Nevertheless, we can't know for sure how progress will extend to other tasks until new approaches emerge for tackling gSCAN.

## Broader Impact

Systematic generalization characterizes human language and thought, but it remains a challenge for modern AI systems. The gSCAN benchmark is designed to stimulate further research on this topic. Advances in machine systematic generalization could facilitate improvements in learning efficiency, robustness, and human-computer interaction. We do not anticipate that the broader impacts would selectively benefit some groups at the expense of others.

## Acknowledgments and Disclosure of Funding

We are grateful to Adina Williams and Ev Fedorenko for very helpful discussions, to João Loula who did important initial work to explore compositional learning in a grid world, to Robin Vaaler for comments on an earlier version of this paper, and to Esther Vecht for important design advice and support. Through B. Lake's position at NYU, this research was partially funded by NSF Award 1922658 NRT-HDR: FUTURE Foundations, Translation, and Responsibility for Data Science.

## Footnotes

[2]`https://github.com/LauraRuis/groundedSCAN`

[3]This means the agent receives a symbolic world representation, not RGB images like in the example figures.

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
