[Supplementary Material · gscan_appendix.pdf]

# A Benchmark for Systematic Generalization in Grounded Language Understanding:
## *Appendix*

**Laura Ruis**[*]
University of Amsterdam
laura.ruis@student.uva.nl

**Jacob Andreas**
Massachusetts Institute of Technology
jda@mit.edu

**Marco Baroni**
ICREA
Facebook AI Research
mbaroni@fb.com

**Diane Bouchacourt**
Facebook AI Research
dianeb@fb.com

**Brenden M. Lake**
New York University
Facebook AI Research
brenden@nyu.edu

## A   The CFG to Generate Input Commands

ROOT $\rightarrow$ VP

| | | |
|---|---|---|
| VP $\rightarrow$ VP  RB | VV$_i$ $\rightarrow$ | {walk} |
| VP $\rightarrow$ VV$_i$ *'to'* DP | VV$_t$ $\rightarrow$ | {push, pull} |
| VP $\rightarrow$ VV$_t$ DP | RB $\rightarrow$ | {while spinning, while zigzagging, hesitantly, cautiously} |
| DP $\rightarrow$ *'a'* NP | NN $\rightarrow$ | {circle, square, cylinder} |
| NP $\rightarrow$ JJ  NP | JJ $\rightarrow$ | {red, green, blue, big, small} |
| NP $\rightarrow$ NN | | |

Where a subscript of 'i' refers to *intransitive* and of 't' to *transitive*.

## B   World State Generation

We generate the world state for each instruction with two constraints: (1) there must be a unique target object for the referent, (2) if a size modifier is present in the instruction, there must be at least one 'distractor' object present. In other words, if the target referent is 'the small square', we additionally place a larger square than the target of any color, and if the target referent is 'the small *yellow* square', we additionally place a larger *yellow* square. For each world generation we select at random half of the possible objects to place, where we make sure that if a size is mentioned, we always put a pair of differently sized objects for each color-shape pair. There are three different situations for which we generate a different set of objects, depending on whether a shape, color and/or size is mentioned.

1. *Shape* (e.g., 'the circle'):
   Generate 1 randomly colored and randomly sized object of each shape that is not the target shape, randomly select half of those. See top-left of Figure 1.

2. *Color and shape* (e.g., 'the red circle'):
   Generate 1 randomly sized object of each color and shape pair that is not the target color and shape. See bottom-left of Figure 1.

3. *Size, color, and shape* (e.g., 'the small circle', 'the red small circle'):
   Generate 2 randomly sized objects for each color and shape pair, making sure the size for

---

[*]Work done during an internship at Facebook Artificial Intelligence Research.

the objects that are the same shape (and color if mentioned) as the target are smaller/larger than the target dependent on whether the size modifier is big/small. Select at random half of the pairs. See top- and bottom-right of Figure 1.

We generate world states for instructions by generating all combinations of possible target objects based on the determiner phrase (e.g., 'the yellow square' gives 4 possible targets, namely a yellow square of each size), all possible relative directions between the agent and the object (e.g., the agent can be to the north-east of the target) and all possible distances between the agent and the object (e.g., if the target is to the north-east of the agent the minimum number of steps is 2 and the maximum is $2 \cdot d$, but if the target is to the north, the minimum is 1 and the maximum $d$). We randomly sample, for each of the combinations, possible positions of the agent and target and this gives us full dataset.

**Command**: walk to the circle while zigzagging.

**Target**: L_turn L_turn walk L_turn walk R_turn walk walk

**Command**: push the red small circle cautiously.
**Target**: L_turn L_turn *L_turn R_turn R_turn L_turn* walk *L_turn R_turn R_turn L_turn* walk *L_turn R_turn R_turn L_turn* walk *L_turn R_turn R_turn L_turn* push

**Command**: walk to the red circle hesitantly.

**Target**: R_turn walk *stay* walk *stay* walk

**Command**: push the small circle while spinning
**Target**: *L_turn L_turn L_turn L_turn* L_turn L_turn walk *L_turn L_turn L_turn L_turn* walk *L_turn L_turn L_turn L_turn* push *L_turn L_turn L_turn L_turn* push

Figure 1: Four real data examples, for a grid size of 6.

## C   Dataset Statistics

Table 1: Number of examples in each dataset. The first two rows for the train and test set denote data for the compositional splits, where GECA denotes the augmented set. An example is unique if it has a different input command, target commands, or target object location. The third column denotes the number of unique world states the agent sees on average per input command.

| **Train** | *Examples* | Unique Examples | |
| --- | --- | --- | --- |
| | | *Total* | *Per Command* |
| Compositional | 367,933 | 76,033 | 177 |
| GECA | 377,933 | 64,527 | 186 |
| Target Length | 180,301 | 80,865 | 599 |
| **Test** | *Examples* | *Unique* | *Per Command* |
| Compositional | 19,282 | 16,381 | 38 |
| GECA | 19,282 | 16,381 | 38 |
| Target Length | 37,784 | 31,000 | 230 |

If we dive a bit deeper in the augmentations to the data GECA makes, it becomes clear why performance deteriorates when this data augmentation technique is used in a dataset like grounded SCAN, and why it is not able to improve performance on the 'yellow squares'-split (Section 5B). In Table 2 we can see that although GECA correctly identifies red square target objects as missing in the training data, and augments data examples to contain target red squares, it is not able to do the same for the instruction command. Even though red squares are now seen as target objects, they are still always referred to without the color (e.g., 'the small square'). See for reference the row with blue squares as targets, which occurs in the non-augmented dataset and is therefore less affected by GECA. Also for yellow squares as target object GECA is not able to add the needed reference to the color. Additionally, when looking at the other row in Table 2, GECA also caused the references to 'red circles' to completely disappear. It seems that it is non-trivial to augment grounded SCAN with GECA to obtain improved performance on anything other but a narrow part of the test set.

Table 2: Some dataset statistics for the compositional splits. In each row the number of examples for a given target object is shown. The column 'placed' means that object was placed in the world as the target, and the column 'referred' means that it was referred to with the color (e.g., 'the red square' or 'the small yellow square')

| | Non-augmented | | Augmented | |
| --- | --- | --- | --- | --- |
| | *Placed* | *Referred* | *Placed* | *Referred* |
| Blue Squares | 33,250 | 16,630 | 16,481 | 5,601 |
| Red Squares | 0 | 0 | 83,887 | 0 |
| Yellow Squares | 16,725 | 0 | 10,936 | 0 |
| Red Circles | 33,670 | 16,816 | 16,854 | 0 |

## D   A Forward-Pass Through the Model

A full forward pass through the model can be represented by the following equations.

**Encoder**

*Command encoder* $\mathbf{h}^c = f_c(\mathbf{x}^c)$: $\hspace{4cm} \forall i \in \{1, \ldots, n\}$

$$\mathbf{e}_i^c = \mathbf{E}_c(x_i^c)$$
$$\mathbf{h}_i^c = \text{LSTM}_{\phi_1}(\mathbf{e}_i^c, \mathbf{h}_{i-1}^c)$$

*State encoder* $\mathbf{H}^s = f_s(\mathbf{X}^s)$:

$$\mathbf{H}^s = \text{ReLU}([\mathbf{K}_1(\mathbf{X}^s); \mathbf{K}_5(\mathbf{X}^s); \mathbf{K}_7(\mathbf{X}^s)])$$

Where $\mathbf{E}^c \in \mathbb{R}^{|V_c| \times d_c}$ is the embedding lookup table with $V_c$ the input command vocabulary, $d_c$ the input embedding dimension, and $\mathbf{K}_k$ the convolutions with kernel size $k$. We pad the input images such that they retain their input width and height, to enable visual attention. That means the world state features are $\mathbf{H}^s \in \mathbb{R}^{d \times d \times 3c_{\text{out}}}$, with $c_{\text{out}}$ the number of channels in the convolutions. We then

decode with an LSTM whose initial state is the final state of the encoder LSTM. The input to the decoder LSTM is a concatenation of the previous token embedding, a context vector computed by attention over the hidden states of the encoder, and a context vector computed by conditional attention over the world state features as processed by the CNN. A forward pass through the decoder can be represented by the following equations.

**Decoder** $p(\mathbf{y}|\mathbf{h}^c, \mathbf{H}^s)$: $\qquad\qquad\qquad\qquad\qquad\qquad\qquad\qquad\quad \forall j \in \{1, \ldots, m\}$

$$\mathbf{h}_0^d = \mathbf{W}_p \mathbf{h}_n^c + \mathbf{b}_p$$
$$\mathbf{e}_j^d = \mathbf{E}^d(y_{j-1})$$
$$\mathbf{h}_j^d = \text{LSTM}_{\phi_2}([\mathbf{e}_j^d; \mathbf{c}_j^c; \mathbf{c}_j^s], \mathbf{h}_{j-1}^d)$$
$$\mathbf{o}_j = \mathbf{W}_o[\mathbf{e}_j^d; \mathbf{h}_j^d; \mathbf{c}_j^c; \mathbf{c}_j^s]$$
$$\hat{\mathbf{o}}_j = p_\theta(y_j \mid \mathbf{x}, y_1, \ldots, y_{j-1}) = \text{softmax}(\mathbf{o}_j)$$
$$\hat{y}_j = arg\max_{V_t}(\hat{\mathbf{o}}_j)$$

*Textual attention* $\mathbf{c}_j^c = \text{Attention}(\mathbf{h}_{j-1}^d, \mathbf{h}^c)$:

$$\qquad\qquad\qquad\qquad\qquad\qquad\qquad\qquad\qquad\qquad\qquad \forall i \in \{1, \ldots, n\}$$

$$e_{ji}^c = \mathbf{v}_c^T \tanh \mathbf{W}_c[\mathbf{h}_{j-1}^d; \mathbf{h}_i^c]$$
$$\alpha_{ji}^c = \frac{\exp(e_{ji}^c)}{\sum_{i=1}^n \exp(e_{ji}^c)}$$
$$\mathbf{c}_j^c = \sum_{i=1}^n \alpha_{ji}^c \mathbf{h}_i^c$$

*Conditional visual attention* $\mathbf{c}_j^s = \text{Attention}([\mathbf{c}_j^c; \mathbf{h}_{j-1}^d], \mathbf{H}^s)$

$$\qquad\qquad\qquad\qquad\qquad\qquad\qquad\qquad\qquad\qquad\qquad \forall k \in \{1, \ldots, d^2\}$$

$$e_{jk}^s = \mathbf{v}_s^T \tanh \mathbf{W}_s[\mathbf{h}_{j-1}^d; \mathbf{c}_j^c; \mathbf{h}_k^s]$$
$$\alpha_{jk}^s = \frac{\exp(e_{jk}^s)}{\sum_{k=1}^{d^2} \exp(e_{jk}^s)}$$
$$\mathbf{c}_j^s = \sum_{k=1}^{d^2} \alpha_{jk}^s \mathbf{H}_k^s$$

Where $\mathbf{W}_c \in \mathbb{R}^{h_d \times (h_e + h_d)}, \mathbf{v}_c \in \mathbb{R}^{h_d}, \mathbf{W}_s \in \mathbb{R}^{h_d \times (3c_{\text{out}} + h_d)}, \mathbf{v}_s \in \mathbb{R}^{h_d}, \mathbf{W}_p \in \mathbb{R}^{h_d \times h_e}, \mathbf{W}_o \in \mathbb{R}^{|V_t| \times (d_e + 3h_d)}$, with $h_e$ the hidden size of the encoder and $h_d$ of the decoder, $c_{\text{out}}$ the number of channels in the encoder CNN, $V_t$ the target vocabulary, and $d_e$ the target token embedding dimension. All model parameters are $\theta = \{\mathbf{E}_c, \phi_1, \mathbf{K}_1, \mathbf{K}_5, \mathbf{K}_7, \mathbf{W}_c, \mathbf{v}_c, \mathbf{W}_s, \mathbf{v}_s, \mathbf{W}_p, \mathbf{E}_d, \phi_2, \mathbf{W}_o\}$.

# E   (Hyper)parameters

**Compute power used**
Training a model on the data used for the experiments with a single GPU takes less than 24 hours.

Table 3: Parameters for the models not explicitly mentioned in Section 4.

| Situation Enc. | Value | Decoder | Value |
|---|---|---|---|
| $c_{\text{out}}$ | 50 | $d_e$ | 25 |
| Dropout $p$ on $\mathbf{H}^s$ | 0.1 | LSTM Layers | 1 |
| **Command Enc.** | **Value** | $h_d$ | 100 |
| $d_c$ | 25 | Dropout $p$ on $\mathbf{E}^d$ | 0.3 |
| LSTM Layers | 1 | **Training** | **Value** |
| $h_e$ | 100 | $\beta_1$ | 0.9 |
| Dropout $p$ on $\mathbf{E}^c$ | 0.3 | $\beta_2$ | 0.999 |
| **GECA** | **Value** | **GECA** | **Value** |
| Gap size | 1 | Maximum gaps | 2 |

# F   Additional Results

This section contains additional experimental results that were referred to in the main text in Section 5.

**Experiment D**

Figure 2: Visualized predictions from the models, where a darker grid means a higher attention weight for that cell.

**Experiment G**

Figure 3: The exact match decreases when target length increases for the adverb split where the agent needs to generalize 'cautiously' after seeing 50 demonstrations. Note that for this experiment, the tested target lengths are *not* longer than encountered during training.

**Experiment H**

Figure 4: The exact match decreases when target length increases for the adverb split where the agent needs to generalize 'while spinning' to the verb 'pull'. Note that for this experiment, the tested target lengths are *not* longer than encountered during training.