[Reviews · NeurIPS 2020]

Review 1

Summary and Contributions: The paper proposes a new benchmark (gSCAN) for learning compositional, grounded natural language understanding and establishes baselines on 8 different tasks.

Strengths: Excellent work building on the SCAN benchmark, exhibiting appropriate understanding of natural language semantics, as well as appropriate ways of evaluating the performance of ANNs on NLU tasks.

Weaknesses: Given the current state of the art of the sub-field, there are no significant weaknesses. Obviously, the formal semantics sub-field of linguistics has systematically investigate many constructions in English and many other languages in a grounded and compositional way, and gscan is still very far from that.

Correctness: Yes.

Clarity: Yes.

Relation to Prior Work: Yes.

Reproducibility: Yes

Additional Feedback:


Review 2

Summary and Contributions: The authors introduce a new benchmark, named gSCAN, for evaluating compositional generalization in situated language understanding. Their main contribution is the design of test sets that require different forms of linguistic generalization.

Strengths: 1. The authors introduce a new benchmark grounded SCAN (gSCAN) and evaluate eight types of compositional generalization in the states of a grid world accessible to the agent. 2. Among the eight different generalization splits, they compare the baseline and GECA methods with sequence matching accuracy as an evaluation metric. It is concluded that it is challenging to account for common natural language generalization phenomena with standard neural models, and that gSCAN could be used as a fruitful benchmark for developing models with more human-like compositional learning skills. 3. It will be of great help to the improvement of the generalization ability of the subsequent models in understanding natural language. This work also has a relatively large impact on the research work of the NeurIPS community in this area.

Weaknesses: 1. Although the author has done a comparative experiment between the multi-modal and the GECA methods on the proposed benchmark, the design and analysis of the experiment are confusing and difficult to understand. 2. Besides, there are lots of inconsistencies in the experiment section, such as the title of Table1 claim that “Models fail on all splits except C and F”, which is inconsistent with the data in rows C and F. The poor performance of the two methods on most split datasets could explain that advances are needed in neural architectures for compositional learning. 3. However, I am not quite sure why the author can draw the following conclusions based on the experiment and analysis of split A-I: artifacts in SCAN are not central to the nature of compositional generalization, gSCAN removes these artifacts by introducing more sophisticated semantics through grounding. I think that seems a little bit unconvincing.

Correctness: The author’s conclusion in the paper seems to be reasonable, and the empirical methodology is also correct in the general direction. However, there are some problems in the experiment design, which make the experiment analysis section difficult to understand, and the experimental results are also unconvincing to support the drawn conclusions.

Clarity: The writing of this paper is good and native, but there are logical inconsistencies, especially in the design and analysis of experiments. This makes it difficult for readers to understand, even if they are relatively familiar with the field.

Relation to Prior Work: The authors mention enough related works in the paper. The most important previous work is related to the SCAN benchmark and Good-enough compositional data augmentation (GECA) method. The author argues that SCAN lacks grounding, which severely limits the variety of linguistic generalizations it can examine. In addition, authors believe that GECA is unreliable in more realistic, grounded language understanding. And the difference between this work and the previous works is also clearly explained.

Reproducibility: Yes

Additional Feedback: 1. I think that the authors should give more detailed description about the grid work of navigation-related tasks, which may be not familiar to most readers. 2. What does SD in the title of the Table 1 mean? 3. What does the <SOS> in the Figure 3 mean? 4. There are some subtle errors in Figure 3, such as the output of Bi-LSTM and Conv. Net should be connected to c_j^c and c_j^s respectively, right? 5. In Table 1, I think the Baseline mean the multi-modal NN method, and GECA corresponds to the previous method, right? Thus, the GECA should be the baseline method, right? 6. Are there some other works which also use the multi-modal NN to model this action sequence prediction task? If so, could you compare to those works? 7. I think language understanding is hard to define for the computer. Can sequence match accuracy be used to measure the ability of language understanding?


Review 3

Summary and Contributions: This paper presents a benchmark for systematic generalization in grounded language understanding, which is an extension from SCAN with grounding to the object knowledge and states. The main contribution of this work is on the multiple new aspects of generalization into the benchmark in addition to the previous configurations. The experimental results show that both the baseline and a SOTA method on the SCAN have limitations in the newly introduced generalization aspects in this proposed benchmark.

Strengths: - This work presented multiple research problems that haven't been addressed by previous work on compositional generalization. - Very detailed discussions were given with the experimental results.

Weaknesses: - There is no significant contribution to the model itself. - The scope of the work is narrow restricted only to the SCAN environment. - Not convinced if the comparisons between just two models are enough to support the main claims of this work.

Correctness: High-level concepts make sense to me.

Clarity: I think this paper could be much further improved in terms of its clarity especially for readers who are not familiar with its previous work. I was also struggling to understand the details before I read the original SCAN paper.

Relation to Prior Work: This work cites many related studies with clear descriptions.

Reproducibility: Yes

Additional Feedback:


Review 4

Summary and Contributions: Summary: the authors propose a new task that focuses on syntactic generalization that is grounded in the states of a grid world. The new benchmark facilitates new development for learning generalization through linguistically motivated rules. Contributions: This paper offers an interesting combination of compositionality and contextuality in natural language understanding. Previous work such as SCAN focuses mostly on compositionality aspect without external grounding to a state of the world. Through a grid world, the authors combine these two aspects for a grounded SCAN benchmark (gSCAN).

Strengths: The proposed benchmark extends the previous SCAN benchmark into a multimodal setting with a grounding state of the world. This new setting is challenging and might render previous methods less effective. For example, the permutation approach proposed in [13] might not be applicable in gSCAN because it would alter the action sequence. gSCAN is, hence, more realistic and requires stronger reasoning and generalization capabilities.

Weaknesses: One weakness of the paper is the synthetic setting of the dataset, e.g. 2D grid world, which is quite unrealistic to real world environment. Another limitation is the fixed hold-out attributes of test splits. For example, split B is limited to test whether the model can ground 'yellow squares' only and split C is limited to 'red squares' only. Therefore, the test results might be affected by data distribution bias in these specific hold-out attributes in the training split.

Correctness: I am a bit confused about the difference between split E and split C. In both splits, at training time, the model does not see a specific novel combination of attributes ("red square" for split C, and "small circle" for split E) and at test time, the model is challenged to ground these attributes. Is the difference between two splits established by color combination (split C) and size combination (split E)? Could you give a potential explanation on why GECA performs well in split C but not in split E?

Clarity: The paper is well written with clear motivation, technical details, and data construction process. The empirical results also show clear details of performance breakdown.

Relation to Prior Work: The authors describe previous work related to compositional generalization tasks. They explained the shortfalls of previous methods when applied into the new benchmark gSCAN with a grounding environment. It would also be useful if the authors can discuss in more detail (probably in a separate paragraph) on work that focuses on contextual generalization (one example would be [8]) and how the new benchmark is different from current work in this line of research. Since grounded/ contextual generalization is an important contribution of the work, similar related work should be highlighted more.

Reproducibility: Yes

Additional Feedback: Is the difference between two splits established by color combination (split C) and size combination (split E)? Could you give a potential explanation on why GECA performs well in split C but not in split E? Web link in Reference [2] is not working.

[Author Response · NeurIPS 2020]

We thank each reviewer for taking the time to thoughtfully comment on our work and we're glad that they recognize
its usefulness and novelty; *R2* says *"It will be of great help to the improvement of the generalization ability of the*
*[...] models in understanding NL."* and *R3* writes that we *"presented multiple research problems that haven't been*
*addressed by previous work [...]."* *R4* says *"gSCAN is [...] more realistic and requires stronger reasoning [...]."*. We
address the concerns and suggestions of 3 reviews below, as *R1* finds *"no significant weaknesses"*.

**The use of synthetic benchmarks (*R3* and *R4*).** For systematic generalization tests we believe it to be important
to limit the scope of the data (*R3*) enough such that one can pinpoint where errors come from. Furthermore, on
discrepancy with real-world environments (*R4*): the issues studied in gSCAN would feature prominently in realistic
NLU tasks, such as teaching autonomous agents to perform tasks by demonstration. gSCAN is distinguished by
evaluating 7 types of compositionality (mostly from a single training set), whereas most benchmarks focus on a single
type. gSCAN's grounding facilitates context-sensitive ("small") and modification-based ("while spinning") forms of
compositionality that go beyond existing tests. That said, since the vocabulary of gSCAN is limited, for future work we
will make it possible to test whether a larger vocabulary (which is trivial extension of the current setup to $N$ object
colors and $M$ shapes) increases performance.

**Response to *R2*.** R2 asks an important question: NLU is hard to define for a computer, can sequence matching
accuracy be used as a proxy for it? We agree with R2 that NLU is hard to define, but we believe that within gSCAN
accuracy is a proxy for understanding. Generating a valid action sequence is only possible if both the command
and world state are understood. Our goal with gSCAN is to be as comprehensive as possible about different kinds
of compositionality, and as controlled as possible about the rest; we don't claim that it is a general-purpose NLU
benchmark. R2 also wonders if gSCAN removes artifacts from SCAN. We are confident of this, as this is precisely
the motivation of designing gSCAN. The methods that are SOTA on SCAN (like [1], [13], [26], [31], [33]) all exploit
artifacts like interchangeability of primitives (e.g., jump, walk, run, look can be used in the same context). In gSCAN,
this is not the case (due to for example the use of adverbs that have a transformative effect on the action sequence
and grounding), and therefore methods that solve SCAN, fail on gSCAN. R2 wonders: don't these results just show
that we need more compositional machinery? We agree with R2, and we hope gSCAN inspires this in the research
community. R2 also points out some things that are unclear in the experiments section. They ask what SD in table 1 and
<SOS> in Figure 3 refer to. Thanks for raising this; SD refers to standard deviation between runs, and <SOS> is the
'start-of-sequence' token that initializes the decoder. On the "lots of inconsistencies in the experiment section", we don't
understand what R2 means; please clarify in the final review so we can address. R2 says the caption is wrong for Table
1, but we can't find a mistake; The claim that the models do not fail on split C and F is based on the higher accuracies
(especially GECA on C). It's true the models also perform well on the random split (A), which we left unsaid but will
add to the caption. Finally, we thank R2 for pointing out 2 missing links in Fig. 3, we will update them accordingly.

**Respose to *R3*.** R3 says a there is no significant contribution to the model, and is concerned that more models are
needed to draw conclusions. The two models we used are representative of many SOTA models and have machinery
that has proven important in NLU (seq. networks, conv. nets, double attention, data aug.). We believe a strength of
gSCAN is that it is currently unclear how to design a model that will perform better on the tasks. We hope gSCAN
sparks more research into compositional architectures that are better at reasoning from little data.

**Response to *R4*.** To clarify the difference between split C and E: C is about generalizing to unseen combinations of
familiar colors and shapes. In the training set red squares are seen as target object, but never referred to with the color
identifier (only "square" or "the big/small square"). Split E asks whether a model can learn that the same expression can
refer to different objects based on the context. R2 wonders why GECA performs well on C, but not on E: the analysis in
the appx. C can shed light on this. GECA adds a lot of red squares to the training set. For split E, GECA doesn't add
anything of use, as it does not identify the issue that the referent "a small circle" is never used for a circle of size 2. A
limitation R4 then points out is: the results might be affected by distribution bias of the fixed held-out attributes. This is
a good point that we thought about. We did an analysis on the distribution of the training set. E.g., yellow squares are
referred to 16,725 times in the training set without the referring expression containing 'yellow' (more of these stats in
the paper Sec. 5). That said, the training distribution indeed still differs from the test distribution, as is often the case in
systematic generalization and few-shot learning: this is what we hope a model can be robust to. Finally, thanks for
pointing out the failing link for [2].

**Clarification about related or previous works.** *R4* wonders how our work differs from work that focuses on con-
textual generalization, like [8]. It indeed seems at first that this is a related paper in terms of the dataset scope. However,
[8] is specifically designed for human-in-the-loop training and sample-efficiency, with no linguistic generalization
involved. R2 and R4 ask us to compare to such related work, so we are happy to add a paragraph highlighting [8] and
other work. *R3* says the paper can be improved for readers unfamiliar with related work. This is useful information
for us, and we are happy to incorporate more background from the original SCAN paper. It would be great if R3 could
clarify which parts are especially unclear and what of the original SCAN paper helped for clearing this up.

[Meta-Review · NeurIPS 2020]

The paper proposes a new benchmark called gSCAN, for learning compositional and grounded natural language understanding. The argument is that to evaluate compositional generalization, situated language understanding (grounding) is necessary. It evaluates eight types of compositional generalization methods with the benchmark. The conclusion is that gSCAN can be used as a useful benchmark. Strength • A new benchmark dataset is created. The work is novel. • The arguments appear to be sound. • Experiments are conducted. Weakness • The paper is generally clearly written. There are places in which the writing can be improved, however. • The setting might be too narrow and artificial from the practical point of view. Discussions have been made among the reviewers. The conclusion is that the paper can be accepted. The authors are strongly suggested to address the writing issues pointed out by the reviewers. It is important to let people in other fields to better understand the content of the paper.